# *PLEK2*, *RRM2*, *GCSH*: A Novel *WWOX*-Dependent Biomarker Triad of Glioblastoma at the Crossroads of Cytoskeleton Reorganization and Metabolism Alterations

**DOI:** 10.3390/cancers13122955

**Published:** 2021-06-12

**Authors:** Żaneta Kałuzińska, Damian Kołat, Andrzej K. Bednarek, Elżbieta Płuciennik

**Affiliations:** Department of Molecular Carcinogenesis, Medical University of Lodz, 90-752 Lodz, Poland; damian.kolat@stud.umed.lodz.pl (D.K.); andrzej.bednarek@umed.lodz.pl (A.K.B.); elzbieta.pluciennik@umed.lodz.pl (E.P.)

**Keywords:** GBM, glioblastoma, WWOX, PLEK2, RRM2, GCSH, biomarkers, cytoskeleton, metabolism

## Abstract

**Simple Summary:**

Cytoskeleton reorganization affects the malignancy of glioblastoma. The *WWOX* gene is a tumor suppressor in glioblastoma and was found to modulate the cytoskeletal machinery in neural progenitor cells. To date, the role of this gene in the cytoskeleton or glioblastoma has been studied separately. Therefore, the purpose of this study was to investigate WWOX-dependent genes in glioblastoma and indicate cytoskeleton-related processes they are involved in. The most relevant WWOX-dependent genes were found to be *PLEK2*, *RRM2*, and *GCSH*, which have been proposed as novel biomarkers. Their biological functions suggest that there is an important link between cytoskeleton and metabolism, orchestrating tumor proliferation, metastasis, and resistance. Searching for such new therapeutic targets is important due to the constant lack of effective treatment for glioblastoma patients.

**Abstract:**

Glioblastoma is one of the deadliest human cancers. Its malignancy depends on cytoskeleton reorganization, which is related to, e.g., epithelial-to-mesenchymal transition and metastasis. The malignant phenotype of glioblastoma is also affected by the *WWOX* gene, which is lost in nearly a quarter of gliomas. Although the role of *WWOX* in the cytoskeleton rearrangement has been found in neural progenitor cells, its function as a modulator of cytoskeleton in gliomas was not investigated. Therefore, this study aimed to investigate the role of *WWOX* and its collaborators in cytoskeleton dynamics of glioblastoma. Methodology on RNA-seq data integrated the use of databases, bioinformatics tools, web-based platforms, and machine learning algorithm, and the obtained results were validated through microarray data. *PLEK2*, *RRM2*, and *GCSH* were the most relevant *WWOX*-dependent genes that could serve as novel biomarkers. Other genes important in the context of cytoskeleton (*BMP4, CCL11, CUX2, DUSP7, FAM92B, GRIN2B, HOXA1, HOXA10, KIF20A, NF2, SPOCK1, TTR, UHRF1,* and *WT1*), metabolism (*MTHFD2*), or correlation with *WWOX* (*COL3A1, KIF20A, RNF141,* and *RXRG*) were also discovered. For the first time, we propose that changes in *WWOX* expression dictate a myriad of alterations that affect both glioblastoma cytoskeleton and metabolism, rendering new therapeutic possibilities.

## 1. Introduction

Amongst gliomas, the most common aggressive primary brain tumor is glioblastoma IDH wild-type (GBM), constituting more than a half of the tumors originating from glia or glial precursors [1,2]. GBM may develop de novo (the primary tumor) or via malignant progression from lower-grade glioma (LGG) as glioblastoma IDH mutated [3]—regardless of derivation, patient’s prognosis is dismal [4]. The malignant phenotype of GBM results from, e.g., dynamics of cytoskeleton [5,6], which has been found to guide signaling pathways [7], proliferation [8], polarity [9], cell cycle [10], or epithelial-to-mesenchymal transition (EMT) [11] and metastasis [12]. Ultimately, cytoskeleton controls two processes that impact on cancer malignant behavior, i.e., cellular division and invasion [6]. Thus, it is not surprising that cytoskeleton and its components have often been considered in anti-cancer therapies or prognosis assessment [12,13,14], including in GBM [6,15,16]. 

One of the genes affecting both cytoskeleton and GBM is WW domain-containing oxidoreductase (*WWOX*), a haploinsufficient tumor suppressor described in many cancers, including GBM, for which it impairs malignant phenotype [17]. In the brain tissue, WWOX can be summarized as a global modulator of transcription and an important regulator of differentiation and maintenance [18]; this is complemented with the prognostic relevance of WWOX [19]. It is also an important modulator of metabolic pathways, regulating the synthesis of amino acids and lipids but also glycolysis or Krebs cycle. Its role in various metabolic disorders [20] or in cancer metabolism [21] was previously reported. Recently, our comprehensive review summarized the current knowledge about *WWOX* in the central nervous system (CNS), including brain tumors, i.e., astrocytoma, neuroblastoma, and glioblastoma [22]. For GBM, it has been found that *WWOX* downregulation may be a result of promoter hypermethylation or loss of heterozygosity (LOH); the latter is related to tumor progression and contributes to 20% of gliomas [22]. Determining the role of *WWOX* in GBM is thought to be in the initial state [19]; hence, profound research is needed, especially in the cytoskeleton-related context, which remains enigmatic. Available data indicate that WW domain of WWOX collaborates, e.g., with dystroglycan, a transmembrane protein that interacts with utrophin and dystrophin, which also communicate with actin [23]. Such finding proves the implication of WWOX in complex machinery involving both extracellular matrix (ECM) and cytoskeleton [23]. Furthermore, it has been found that WWOX recognizes the PPxY motif of ezrin [24], a protein that links transmembrane signaling to the reorganization of cytoskeleton [25], influencing cell migration and tumor progression [24]. In the case of brain tissue, the ability of WWOX to bind and regulate glycogen synthase kinase 3β (GSK-3β) was proved to be important for Alzheimer’s disease [26], in which a frequent *WWOX* downregulation is found among patients [27]. Lastly, *WWOX* silencing changed the expression profile of genes (e.g., *DCLK*, *NEFL*, *NEFM*, *MAP2/4/6*) involved in the cytoskeleton organization of neural progenitor cells [18].

It can be assumed that as a global modulator, *WWOX* will manifest its function directly or indirectly. Therefore, this research aims to examine the role of *WWOX*-dependent genes in terms of cytoskeleton reorganization in GBM. Simultaneously, the lethal nature of the de novo IDH wild-type glioblastoma does not leave too many solutions, and hence it is justified to investigate not only GBM but also low-grade tissue from which glioblastoma IDH mutated could arise afterwards.

## 2. Results

### 2.1. WWOX Influenced a Number of Pathways and Biological Processes

Optimal *WWOX* expression cut-point was determined to separate high- and low-expressing groups of patients. The obtained cut-off value, 222.6, had significantly separated groups, showing better prognosis for patients with high *WWOX* expression level (Figure 1).

Performing gene set enrichment analysis (GSEA) with the use of the selected cut-point confirmed the presence of gene sets such as BIOCARTA/KEGG/REACTOME canonical pathways; chemical and genetic perturbations (CGP); gene ontology (GO): biological processes (BP) and molecular function (MF); cancer gene neighborhoods (CGN); and oncogenic and immunologic signatures, hallmarks, and positional gene sets. Collectively, 74 gene sets were taken into account, which was equivalent to 3046 core enrichment genes (Appendix A). Ultimately, duplicate removal yielded 1898 genes.

### 2.2. Global Distinction Indicated Both Clinical Features and Modules Worth Considering

Together with their corresponding normal brain tissue, the GBM and LGG cohorts were distributed across Uniform Manifold Approximation and Projection (UMAP) dimensions using the 1898 genes from GSEA. Out of eight characteristics acquired from The Cancer Genome Atlas (TCGA) clinical data, half showed promising distribution. First, samples were divided on the basis of the *tumor_type*; normal brain samples were also included to allow simultaneous visualization with tumor samples using one variable (Figure 2A). Subsequently, samples were grouped according to *neoplasm_histologic_grade* (Figure 2B) and *histological_type* (Figure 2C), and finally *age_at_initial_pathologic_diagnosis* divided by median age (Figure 2D). The remaining clinical features (excluded from further analyses due to the lack of a clear pattern) are presented in Appendix A.

The heatmaps were elaborated to split 1898 genes into modules and ease interpretation of specific comparison. The content of the modules is summarized in Appendix A. Representative modules were selected on the basis of the contrasting pattern between the groups and the level of change in expression. For *tumor_type* (Figure 3A), the GBM was distinguished from normal brain (abbreviated NT) using modules 5 and 6. Module 1 was chosen to explain the differences between LGG and NT. GBM and LGG demonstrated different patterns for all modules; hence, two sets were adopted for independent examination—the first set comprised modules 2, 4, 5, and 7 combined together, while the second contained modules 1, 3, and 6. For *neoplasm_histologic_grade* (Figure 3B), the G4 vs. G2 or G4 vs. G3 comparisons were explained using sets of modules 2 + 4 + 5 + 7 and 1 + 3 + 6. Regarding *histological_type* (Figure 3C), we focused on the most drastic changes: the set of modules 1 + 5 + 7 allowed astrocytoma (A) to be distinguished from both oligoastrocytoma (OA) and oligodendroglioma (OD). Additionally, using combined modules 1, 4, 5, and 6, the following comparisons were also included to identify treatment-related differences: (1) *glioblastoma_multiforme* combined with *untreated_primary_gbm* vs. *normal_nervous_system*; (2) *glioblastoma_multiforme* combined with *untreated_primary_gbm* vs. *treated_primary_gbm*; (3) *treated_primary_gbm* vs. *normal_nervous_system*. Lastly, for *age_at_initial_pathologic_diagnosis*, we indicated two sets of modules (1 + 3 and 2 + 4 + 5) that demonstrated opposite expression profile (Figure 3D). Henceforth, sets will be referred to as module numbers linked with “+”.

### 2.3. Differentially Expressed Genes Were Identified for Each Comparison 

Each comparison with corresponding set was summarized through volcano plot. The control in specific comparison was always the phenotype more resembling the natural/normal conditions, e.g., for high-grade vs. low-grade, the latter was used as a reference. A dedicated log_2_ fold-change (log_2_FC) was applied to specific analysis since the number of genes in a given comparison varied depending on the chosen modules. Using a different cut-off log_2_FC value, it was possible to indicate a small but strict group of genes presenting remarkable difference in expression. Differentially expressed genes (DEGs) were considered significant when at least log_2_FC = 0.6 (i.e., fold-change ≈1.5) and *p* < 0.01; however, in many comparisons, even log_2_FC = 3.5 was used to provide the most stringent group of DEGs.

For GBM vs. LGG comparison, 16 DEGs were identified through 1 + 3 + 6 and 2 + 4 + 5 + 7 sets; of these, 11 genes were upregulated and 5 were downregulated. The 5 + 6 set of GBM vs. NT comparison indicated 7 DEGs with log_2_FC = 1 (3 up- and 4 downregulated), while module 1 of LGG vs. NT comparison determined 19 genes having log_2_FC > 3.5, with the vast majority being upregulated. The results for *tumor_type* are presented in Figure 4A. Secondly, three comparisons for *neoplasm_histologic_grade* (G4 vs. G2; G4 vs. G3; G3 vs. G2) were visualized through the 1 + 3 + 6 and 2 + 4 + 5 + 7 sets, resulting in six independent graphs. Altogether, more than 40 DEGs were selected, as summarized in Figure 4B. Subsequently, the 1 + 3 set indicated four DEGs between groups of *age_at_initial_pathologic_diagnosis* (Figure 4C). Concerning histological types, astrocytoma differed from both oligoastrocytoma and oligodendroglioma in terms of only three genes of set 1 + 5 + 7 with log_2_FC > 2.5. The remaining comparisons for this clinical feature revealed several dozen DEGs between the specific GBM phenotype vs. normal brain, and a few ones between treated (Tr.) or untreated (Untr.) high-grade gliomas based on the 1 + 4 + 5 + 6 set (Figure 4D).

## 3. The Most Relevant Genes Were Subjected to ROC Analysis

Sequential elimination using Multiple Support Vector Machine Recursive Feature Elimination (mSVM-RFE) was performed to select the best three features (genes) per comparison. Due to small size of the groups, all comparisons for *histological_type* were based on the topTags() function. Ultimately, 14 out of 16 comparisons contained DEGs; these revealed 42 genes with supposed potential. However, some genes were identified in more than one comparison, resulting in a group of 30 genes. These were henceforth described as *top genes*. The most relevant genes, along with their ascending average rank (AvgRank) or with topTags() hierarchy, are listed in Table 1. 

Elimination of insignificant or overfitted genes was possible via determination of area under the curve (AUC) in receiver operating characteristic (ROC) analysis. Some comparisons contained no valuable predictor, and data validation helped to decide whether to exclude them from subsequent step. The ROC analysis for all top genes is collected in Figure 5. The importance of top genes regarding glioma or cytoskeleton regulation is summarized in Appendix A.

### 3.1. Validation Verified the Usefulness of Top Genes

Independent microarray cohorts were used to certify RNA-seq findings; identical top genes were considered in specific clinical comparison (Figure 6).

This allowed us to exclude 3 out of 14 comparisons from further investigation; these comparisons were G3 vs. G2, set 1 + 3 + 6; A vs. OA + OD, set 1 + 5 + 7; untreated vs. treated GBM, set 1 + 4 + 5 + 6. For the remaining comparisons, one gene with the highest mean AUC was selected. This yielded 9 genes from 11 significant comparisons, as the *NF2* gene was appointed three times. These were *C15orf48*, *CMTM6*, *GCSH*, *HOXA1*, *NF2*, *PLEK2*, *RRM2*, *SAA2*, and *UHRF1*; together with their means of AUC, 95% confidence interval (CI), and accuracy (ACC), they are presented in Table 2.

### 3.2. Several Substantial Genes Were Correlated with WWOX and Possessed Prognostic Value

Prior to focusing on the best nine explanatory genes, we correlated all top genes with *WWOX* to select subsidiary genes more related to *WWOX* but with slightly worse predictive properties. As forecasted, among all top genes (Table 3), and besides nine predictors, there were a few additional genes worth considering, e.g., *COL3A1*, *KIF20A*, *RNF141*, and *RXRG*. This resulted in a total of 13 genes that were then subjected to survival analysis, together with *WWOX* (Figure 7). Kaplan–Meier curves revealed that despite being at the forefront, *UHRF1* was not found to be of prognostic importance and was excluded (Figure 7N).

### 3.3. PLEK2, RRM2, and GCSH Were the Most Valuable in Differentiating Gliomas and Correlated with WWOX

The remaining 12 genes were visualized in terms of their expression across all RNA-seq samples. The selection of three the most relevant genes was based on their predictive value (AUC), effect on survival (DFS), correlation with *WWOX*, and transparency of expression difference across UMAP dimensions. These were found to be *PLEK2*, *RRM2*, and *GCSH*. Together with *WWOX*, their expression difference across UMAP is visualized in Figure 8; the same for the remaining genes can be found in Appendix A. 

The GBM cohort demonstrated markedly increased *PLEK2* and lowered *GCSH* expression compared to LGG, as well as a change in *RRM2* expression similar to that of *PLEK2*. According to Gene Expression Profiling Interactive Analysis (GEPIA2), the last two are positively correlated, i.e., R = 0.62, *p* < 0.0001 (Figure 9A); a negative correlation can be seen for *PLEK2* and *GCSH*, i.e., R = −0.32, *p* < 0.0001 (Figure 9B). Lastly, *GCSH* and *RRM2* are not well correlated, indicated by a low correlation coefficient, i.e., R = −0.23, *p* < 0.0001 (Figure 9C). Nevertheless, each of the three genes significantly correlated with *WWOX* (*GCSH* positively, *PLEK2* and *RRM2* negatively) as shown in Table 3. Finally, the prognostic signatures of *PLEK2*, *RRM2*, and *GCSH* were adjusted using survival curves with “Estimation of Stromal and Immune cells in Malignant Tumors using Expression data” (ESTIMATE) score as covariate (Figure 9D–F).

## 4. Discussion

Glioblastoma is one of the deadliest human cancers [28]. Its malignant behavior results from, e.g., cytoskeleton reorganization that controls EMT and metastasis [11,29]. GBM is also affected by WWOX, a cytoskeleton-related protein that interacts with ezrin, dystroglycan, or GSK−3β; it also influences on expression of *DCLK*, *NEFL*, *NEFM*, and *MAP2/4/6* genes. As a tumor suppressor, it is lost in nearly a quarter of gliomas through a LOH event [22,23,26]. This study is the first to evaluate the role of *WWOX* in cytoskeleton dynamics of glioblastoma. Low-grade glioma (from which IDH-mutated GBM could arise) was included in the analysis to determine whether *WWOX* loss might also switch expression profile at the earlier stages of carcinogenesis; this may potentially lead to an improved on-time diagnosis.

The main research milestones were (1) acquisition of a list of core enrichment genes using Evaluate Cutpoints and GSEA; (2) selection of the most important clinical features and comparisons using Monocle3; (3) establishment of the top genes and determining their role in the context of glioma and/or cytoskeleton; (4) validation of the results and selection of the three most relevant genes based on their predictive properties (AUC), impact on patients survival (DFS), correlation with *WWOX*, and transparency of spatial expression analysis across UMAP dimensions. Performing the edgeR → mSVM-RFE → ROC workflow independently for each clinical feature is justified, since the final verdict (indicating the most relevant gene) depended on the analyzed trait. Moreover, additional correlation analysis between the top genes and *WWOX* identified four important genes; likewise, multiple testing of the top genes allowed us to spot overfitted genes.

RNA-seq data showed that patients’ prognosis worsens with a lower level of *WWOX*, which is consistent with previous findings in colon, bladder, breast [30], ovarian, prostate [22], gastric, cervical [31], and non-small cell lung [32] cancers. GSEA revealed that WWOX phenotype alters many pathways, e.g., signaling of tumor necrosis factor alpha (TNF-α) via nuclear factor kappa B (NFκB), mammalian target of rapamycin complex 1 (mTORC1), phosphoinositide 3-kinase/protein kinase B/mTOR (PI3K/AKT/mTOR), protein 53 (p53), or Janus kinase 2/signal transducer and activator of transcription (JAK/STAT). The other signatures also revealed hallmark, oncogenic, or computational gene sets; of these, many were found in the literature as *WWOX*-dependent processes (EMT [33], inflammatory response [34,35]) or pathways (TNF-α/NFκB [36], PI3K/AKT/mTOR [37], p53 [38], JAK/STAT [39]). The link between EMT and the cytoskeleton elements is necessary for mechanical strength and structural design involved in cancer migration and invasion [11,29]. Furthermore, the cytoskeleton was found to be crucial for inflammation-related processes, e.g., migration, cytokine production regulation, cell signaling, and adhesion [40]. Regarding pathways, TNF-α/NFκB [41], PI3K/AKT/mTOR [42], p53 [43], and JAK/STAT [44] have been implicated in cytoskeleton remodeling.

Our study revealed a noticeable subdivision of samples based on tumor type (or accompanied normal tissue), grading, histological type, or age. It was foreseeable that a higher grade of LGG (grade 3) or the presence of an astrocytoma histology are more indicative of GBM than lower grade or other types, as IDH mutated glioblastomas derive from astrocytomas [45]. However, it is not possible to make a complete distinction from oligodendroglioma as there are some glioblastomas with oligodendroglioma component (termed GBM-O) that might be pathologically defined as anaplastic oligoastrocytomas with necrosis [46,47]. The mean age has been also included as one of the patterns distinguishing glioblastomas [48], and the clinical practice confirms that application of surgical resection, radiotherapy, or chemotherapy is hampered in older age [49]. 

According to DEG investigation, we established the “top genes” term that contains 30 genes. More than half of them were found to be implicated in the regulation of cytoskeleton (i.e., *BMP4, CCL11, CUX2, DUSP7, FAM92B, GRIN2B, HOXA1, HOXA10, KIF20A, MTHFD2, NF2, PLEK2, RRM2, SPOCK1, TTR, UHRF1, WT1),* influencing actin, vimentin, ezrin, radixin, moesin, or tubulin. Most of the top genes were upregulated and/or contributed to promotion of GBM tumorigenicity, but few of them were downregulated and/or identified as protective prognostic factors, e.g., *BMP4, DUSP7, GCSH, NF2, PHF5A,* and *RXRG*.

At the post-validation stage, there were nine genes with at least acceptable discriminatory properties (AUC > 0.7) [50]. Most have previously been proposed as biomarkers in a specific tumor (or tumor-related condition): *C15orf48* in esophageal cancer [51], *NF2* in neurofibromatosis [52], *CMTM6* in renal adenocarcinoma [53], *PLEK2* in lung adenocarcinoma [54], *RRM2* in glioma [55], *HOXA1* in breast and gastric cancers [56], and *SAA2* in renal cancer [57]. Interestingly, *UHRF1* was indicated as a universal cancer biomarker [58], while *GCSH* seems to be newly observed. According to the modern algorithm developed by the Alliance of Genome Resources [59], the “automated description” indicates that GCSH is biomarker of amino acid metabolic disorder, which is a known abnormality in GBM [60,61]. Four additional genes were also included, as they presented high correlation with WWOX; these were *COL3A1, KIF20A, RNF141*, and *RXRG. COL3A1, KIF20A*, and *RXRG* were described as valuable prognostic biomarkers in ovarian, renal cell, and breast carcinomas, respectively [62,63,64]. DFS analysis of selected 13 genes showed that *GCSH, NF2, RNF141,* and *RXRG* prolong survival while *C15orf48, CMTM6, COL3A1, HOXA1, KIF20A, PLEK2, RRM2*, and *SAA2* shorten it.

The final stages of this study involved the selection of the best three genes—these were *PLEK2*, *RRM2*, and *GCSH*. They correlated with *WWOX*, and favorably (*GCSH*) or unfavorably (*PLEK2* and *RRM2*) affected DFS. Across UMAP dimensions, *GCSH* was downregulated in GBM compared to LGG, but *PLEK2* and *RRM2* were upregulated. Finally, *RRM2* and *PLEK2* were correlated with each other, as were *GCSH* and *PLEK2,* but not *RRM2* and *GCSH*; all three genes were also prognostic when ESTIMATE covariate was applied. Regarding phenotype comparisons for which these genes were representative, both *GCSH* and *PLEK2* distinguished cancer grades: *GCSH* expression was lower in G4 than in G3, while *PLEK2* was higher in G4 compared to G2. The changes in cytoskeleton were linked to increasing cancer grade, e.g., in colon cancer or glioblastoma. At first, Pachenari et al. reported that the proportion of actin and tubulin differs between low and high grade and that microtubules reorganize cytoskeleton to facilitate benign to malignant phenotype transition [65]. Secondly, Reiss-Zimmermann et al. found G4 glioblastoma to be softer than G3 astrocytoma [66]; this stems from cell stiffness, which is related to cytoskeleton reorganization [67]. On the other hand, *RRM2* discriminated untreated GBM from normal nervous system; this is embedded in the TCGA clinical data of *histological_type* but it can also be considered as a typical comparison between glioblastoma and normal brain tissue. As previously mentioned, *RRM2* has been already proposed as an overexpressed biomarker with functional significance in glioma [55]. Hence, *PLEK2* and *GCSH* should also be considered as prognostic biomarkers in GBM, since their spatial expression analysis was even clearer than that of literature-supported *RRM2*.

To understand whether these three genes can undergo targeted therapy, it is necessary to analyze their biological function. In addition, *GCSH* expression is clearly lowered with increasing glioma grade, while that of *PLEK2* and *RRM2* is clearly elevated; therefore, *GCSH* does not fit the premise of targeted therapy, i.e., biological pathway inhibition [68]. Additionally, *GCSH* encodes one of four proteins responsible for glycine metabolism, which is only a single pathway of the whole complicated metabolic machinery [69]; its alteration should be considered in a broader perspective. Complexity is even greater as the cytoskeleton is implicated in carbohydrate metabolism [70], and conversely, actin and tubulin require energy from nucleotide hydrolysis to maintain structural dynamics [71]. Moreover, the other top gene, i.e., *MTHFD2*, was found to regulate metabolism through the folate cycle [72]. It was the best explanatory gene for the differences between LGG and normal brain tissue, suggesting that metabolic changes are not only restricted to glioblastoma.

*RRM2* encodes β subunit of ribonucleotide reductase (RR), an enzyme acquiring 2′-deoxyribonucleotides from ribonucleotide 5′-diphosphates, which are crucial for the synthesis or repair of DNA [73]. It forms a dimer that may bind DNA [74]—each RRM2 monomer contains the tyrosyl radical and non-heme iron [73]. A sufficient supply of deoxyribonucleotides is required for uncontrolled DNA replication in cancer [75]; it is therefore not surprising that *RRM2* is often a target of molecular therapy [73,76,77]. Nowadays, there are several RRM2 inhibitors, i.e., radical scavengers, iron chelators, subunit polymerization inhibitors, or expression silencers [76,78,79,80]. A wide range of anti-RRM2 approaches focus on inhibition of proliferation, differentiation, and division but also invasion [75]. It has been found that *RRM2* knockdown inhibits cell proliferation via DNA damage-driven senescence induction [77].

Considering PLEK2, a protein with two pleckstrin homology (PH) domains and one disheveled-Egl10-pleckstrin (DEP) domain [81] that is able to bind acidic phospholipids of cell membranes [82] or influence actin dynamics [83], targeted therapy is less developed yet already viable. Han et al. screened for small molecules potentially able to bind PLEK2 and identified compounds binding DEP domain; the lead compound, i.e., NUP−17d diminished proliferation similarly to ruxolitinib [82]. Despite moving to the cell membrane through its ability to bind phosphoinositides [84], it is also the effector of increased proliferation driven by JAK/STAT and PI3K/AKT signaling [82,83]. These two pathways were found in GSEA during discrimination of *WWOX* high/low groups. For JAK/STAT, Zhao et al. performed quantitative PCR preceded by chromatin immunoprecipitation (ChIP) to confirm the presence of STAT5 consensus-binding sites in *PLEK2* promoter region [83]. Likewise, PLEK2 recruits phosphatidylinositol 3,4-bisphosphate and proteins, e.g., AKT, phosphoinositide-dependent kinase−1 (PDK1), PDK2, and mTOR, forming a complex that augments PI3K signaling [82]. Since *PLEK2* induces cell spreading and guides tumor progression and metastasis [81], the development of new targeted therapy aimed at this oncogenic molecule is of utmost importance.

Finally, GCSH (or H-protein) is an integral core protein of glycine cleavage system (GCS), the major pathway of glycine degradation [85]. In short, GCS consists of a four-protein complex that catalyzes glycine degradation into carbon dioxide, ammonia, and a methylene group that is accepted by tetrahydrofolate [86]. The functioning of GCS is reversible yet requires an intermediary state when the ternary complex is formed, i.e., compound of P-protein (or glycine decarboxylase; GLDC), the aminomethyl moiety of glycine, and H-protein [87]. This proves that in order to maintain glycine synthesis/degradation for subsequent processing, GCSH must be intact. Here, our study indicates that *GCSH* expression dramatically declines with increasing glioma grade. This is even more intriguing since cancer may benefit from GCS, as stated by Zhang et al. during lung adenocarcinoma research, wherein overexpression of *GLDC* increased tumor formation [88]. However, activity of GCS is mainly restricted to normal tissues of the brain, liver, and kidney [89], which suggests that context is tissue-dependent. Moreover, the brain is one of the few tissues with naturally high *GLDC* expression [90], suggesting the inverse tendency in typically non-GCS-expressing tissues that become cancerous and then overexpress *GLDC*. This can be supported by the findings of Zhuang et al. on hepatocellular carcinoma (HCC), wherein restoration of GCS proper activity (via *GLDC* overexpression due to its low level in malignant HCC) suppressed cancer progression via inhibition of both invasion and metastasis [91]. As mentioned above, the liver is (alongside the brain) one of few tissues maintaining the activity of GCS, and hence GLDC and other proteins of this complex. Therefore, this complex may be inhibited during carcinogenesis; this is in contrast to tissues with a naturally low or absent activity of GCS, where it would be increased. The main purpose of such a switch remains elusive, but to investigate this, glycine metabolism must be considered together with serine as they both are biosynthetically linked; they contribute to the one-carbon metabolism that cycles units of carbon from various amino acids [92]. Many discrepancies around glycine function have arisen, i.e., its uptake has been correlated with cancer proliferation [93], while excess glycine in the diet inhibited tumorigenesis in vivo [94,95]. This is supposedly due to the overall complexity of the metabolism but also insufficient understanding of glycine metabolism in carcinogenesis [96]. In theory, glycine is able to provide all precursors required to support nucleic acid synthesis [97], which is crucial for maintaining cancer cell growth [92]. Interestingly, serine is the main donor of one-carbon units [96] and a central hub of metabolic pathways in cancer [92], which might suggest its supremacy over glycine. Labuschagne et al. report that serine, rather than glycine, supports carbon metabolism and later proliferation [97]. The same authors highlighted that many tumors prefer serine consumption over glycine, and the latter does not substitute serine in nucleotide synthesis; what is more, the paradox of high glycine uptake during rapid proliferation can be a consequence rather than a cause of an such event [97]. This corresponds to our results, suggesting that GBM may preferably switch to serine consumption via *GCSH* downregulation. As *GCSH* was the gene that can distinguish G4 from G3 brain tumors, we speculate this occurs (beyond IDH wild-type GBM development) over the transformation from G3 astrocytoma to G4 glioblastoma. The only concern is how such deadly and incurable tumor handles with excess glycine, which if not metabolized may be converted to toxic by-products such as aminoacetone or methylglyoxal [86]. The explanation may again be serine-dependent; it has been found that not only excess glycine drives conversion to serine and inhibits glycine flux to purines but also high serine leads to glycine efflux [97]. Moreover, GBM is thought to be adapted to environmental conditions via serine-dependent redox homeostasis that enhances tumor survival [98]. Although glioblastoma metabolism remains poorly understood, it appears that antimetabolic therapy focused on serine pathway inhibition may be worth considering.

## 5. Materials and Methods

### 5.1. Data Collection of Cancer Patients and Cut-Point Determination

The entire methodology is summarized in Figure 10. Expression data of RNA-seq together with corresponding clinical annotation were collected from both GBM and LGG cohorts of TCGA-dedicated FireBrowse Repository (level 3 RNA-seqV2, RSEM normalized, data version of 28 January 2016 available at http://firebrowse.org/, accessed on 10 December 2020). Patients missing expression or clinical data were discarded from the study; no additional exclusion criteria were applied. The available data of paired normal brain tissues were additionally retrieved via the R-dedicated package TCGA-Assembler [99]. A total of 672 samples were included in the study. Determining the suitable *WWOX* expression cut-point to stratify the population into two groups was achieved with the help of the R-based Evaluate Cutpoints tool designed in our Department [100].

### 5.2. Identification of Significant Differences between Phenotypes

In order to determine which genes are WWOX-related in high-grade glioma, we conducted GSEA (https://www.gsea-msigdb.org/gsea, accessed on 20 December 2020) on eight major collections acquired from http://software.broadinstitute.org/gsea/msigdb (accessed on 20 December 2020). Functional analysis was performed on 20,502 genes using the tTest metric with a weighted statistic to score hits/misses and permutation type concerning phenotype. From the whole and excluding duplicates, 1898 genes belonging to selected gene sets were chosen with a significance threshold of FDR < 0.25.

### 5.3. Global Profiling and Determination of Gene-Containing Modules

Phenotype heterogeneity between GBM/LGG and normal brain tissue was investigated using the Monocle3 R package [101]. The dimensionality of the reduced space of 100 (num_dim) was chosen for the PCA pre-processing step (preprocess_cds()). Dimension reduction (reduce_dimension()) and individual clustering (cluster_cells()) within spaces were applied with the UMAP algorithm for dimensionality reduction method, upon which they were applied to base clustering (reduction_method). The clusters of individuals were compared with graph_test() function in accordance with Moran’s I spatial autocorrelation analysis with knn neighbor graph and 0.05 q-value. Moreover, the genes varying across the clusters were grouped into modules through Louvain community analysis (find_gene_modules()) with parameters set to default. The modules were clustered on all of the individuals, enabling simultaneously comparison between tumors (e.g., GBM vs. LGG) and tumor vs. non-tumor samples (e.g., GBM/LGG vs. NT). The results were clustered with the Ward D2 method and visualized with pheatmap(). Expression differences for a given gene in the global projection were visualized using plot_cells(). The whole pipeline was performed according to the Monocle3 tutorial (https://cole-trapnell-lab.github.io/monocle3/, accesssed on 26 December 2020).

### 5.4. Examination of Differentially Expressed Genes

Bioconductor’s edgeR package allowed us to find DEGs using embedded DGEList() constructor and fitting a negative binomial generalized log-linear model to the read counts for each gene through glmFit() and glmLRT() with subsequent makeContrasts() between groups of specific comparison (default parameters). The most differentially expressed genes were extracted in the form of the table using topTags() ranked by absolute logFC with *p* < 0.01 adjusted using Benjamini and Hochberg correction method. DEGs were visualized on volcano plots using ggrepel package and geom_text_repel() function.

### 5.5. Relevant Features Investigation

Multiple SVM-RFE approach by Duan et al. [102], which extended techniques of resampling compared to the original idea invented by Guyon et al. [103], was used to obtain top features (genes) across folds. Feature ranking was performed using svmRFE() function with k-fold cross validation (CV) of k = 10 to include a multiplicity of mSVM-RFE and halve.above = 100. R-package *e1071* was included in the environment to allow SVM fitting. After setting up 10-fold CV, we performed feature ranking on all training sets and obtained top features across all folds using WriteFeatures() with a list of genes ordered by ascending AvgRank value (the lower number the better). The best genes (three per each significant comparison) were subjected to further investigation.

### 5.6. Evaluation of Statistical Model Accuracy

The ROC curves were evaluated on genes acquired from SVM machine learning algorithm. The pROC package was used for ROC analysis (estimating AUC, 95% CI, ACC) and for plotting curve plot through ggroc using ggplot2 in R environment.

### 5.7. Validation of the Results

Validation of the findings on the basis of independent adequate cohorts was performed through Agilent 244K G4502A microarray normalized expression data acquired from the University of California Santa Cruz (UCSC) Xena repository. No LGG cohort exists on other platforms, e.g., AffyU133a (GBM microarray data alone was not sufficient to reflect comparisons).

### 5.8. Correlation Analysis and Survival Curves

GEPIA2 (http://gepia2.cancer-pku.cn, accessed on 29 December 2020) was used to correlate the top genes with *WWOX* using Spearman’s rank correlation coefficient. Analysis of genes’ prognostic models on DFS endpoint in both GBM and LGG cohorts (merged together) was conducted using survminer and forestmodel R-packages. Survival curves were set on median group cut-off and forest plots were generated with Cox proportional hazards model. For the best three genes selected from the study, we developed the models with confounder-adjusted survival rate via inverse probability weighting (IPW) with the use of RISCA and hrIPW R-packages. The covariate was ESTIMATE combined score [104], which was divided into high and low groups on the basis of the median value. The log-rank *p*-value was calculated using ipw.log.rank() function of RISCA, while hazard ratio estimation was acquired from hrIPW.

## 6. Conclusions

The change in *WWOX* gene expression dictates a myriad of alterations through *WWOX*-dependent genes that affect both glioblastoma cytoskeleton and metabolism. The greatest differences appear between glioblastoma and other gliomas (particularly in relation to tumor grade G4 vs. G3/G2), where *GCSH* and *PLEK2* can be used to distinguish them since they exhibit opposite expression profiles in this regard. Together with the already described glioma biomarker, *RRM2*, we considered *PLEK2* and *GCSH* as a novel *WWOX*-dependent biomarker triad of glioblastoma, whose subsequent investigation is advisable. *RRM2* and *PLEK2* appear to be prognostic and therapeutic biomarkers; modulating their activity through targeted therapy might inhibit uncontrolled DNA replication or metastasis and proliferation, respectively (presumably driven by JAK/STAT and PI3K/AKT). Regarding *GCSH*, targeted therapy is not justified as its expression decreases with increasing glioma grade; hence, it is not a therapeutic biomarker. However, a broader view on one-carbon metabolism implied that more emphasis should be given to the serine pathway as a possible antimetabolic target. Nevertheless, the usefulness of *PLEK2*, *RRM2*, and *GCSH* as diagnostic or predictive biomarkers is yet to be confirmed. Other genes may also be worth investigating, including those playing a key role in the cytoskeleton (*BMP4, CCL11, CUX2, DUSP7, FAM92B, GRIN2B, HOXA1, HOXA10, KIF20A, NF2, SPOCK1, TTR, UHRF1,* and *WT1*), metabolism (*MTHFD2*), or correlation with *WWOX* (*COL3A1, KIF20A, RNF141,* and *RXRG*).

## Figures and Tables

**Figure 1 cancers-13-02955-f001:**
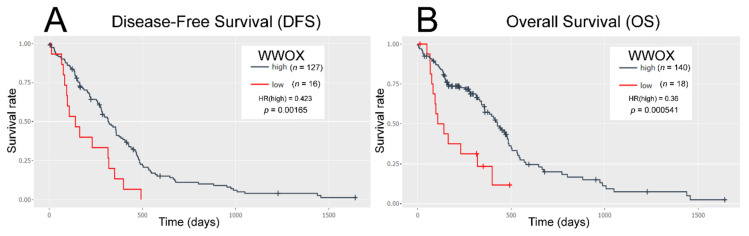
Survival analysis of low and high *WWOX* expression cut-off through Evaluate Cutpoints. (**A**) Disease-free survival (DFS). (**B**) Overall survival (OS).

**Figure 2 cancers-13-02955-f002:**
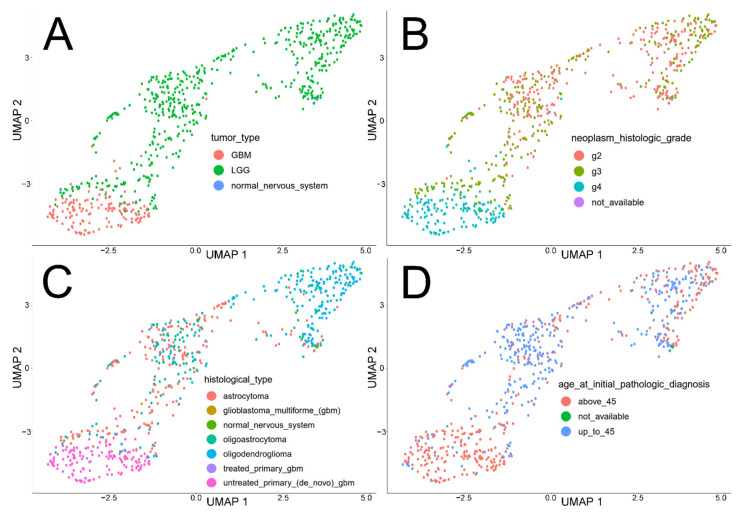
Global profiling of TCGA glioma samples with normal brain tissue through various clinical data. (**A**) *tumor_type* (specific cancers and corresponding normal tissue). (**B**) *neoplasm_histologic_grade*. (**C**) *histological_type*. (**D**) *age_at_initial_pathologic_diagnosis*.

**Figure 3 cancers-13-02955-f003:**
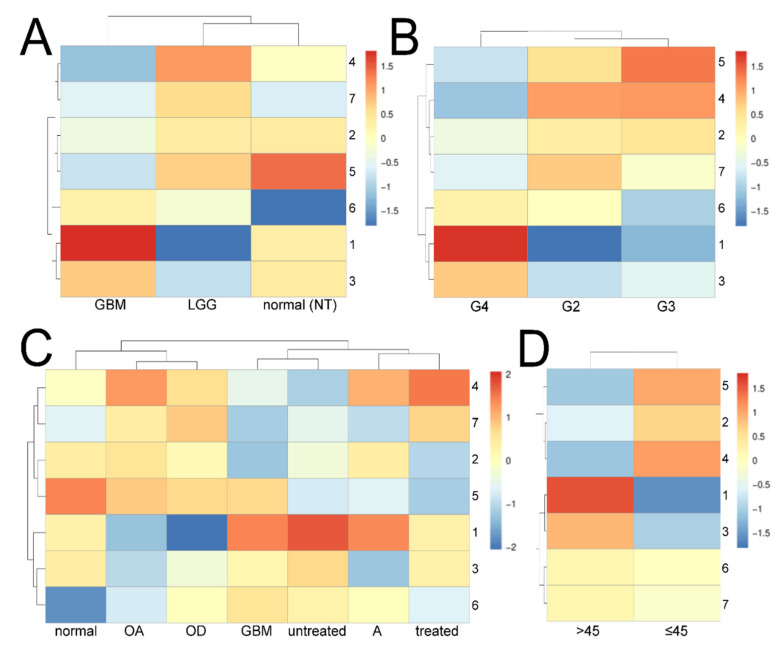
Heatmaps differentiating gliomas and normal brain tissue. (**A**) *tumor_type* (specific cancers and corresponding normal tissue). (**B**) *neoplasm_histologic_grade*. (**C**) *histological_type*. (**D**) *age_at_initial_pathologic_diagnosis*.

**Figure 4 cancers-13-02955-f004:**
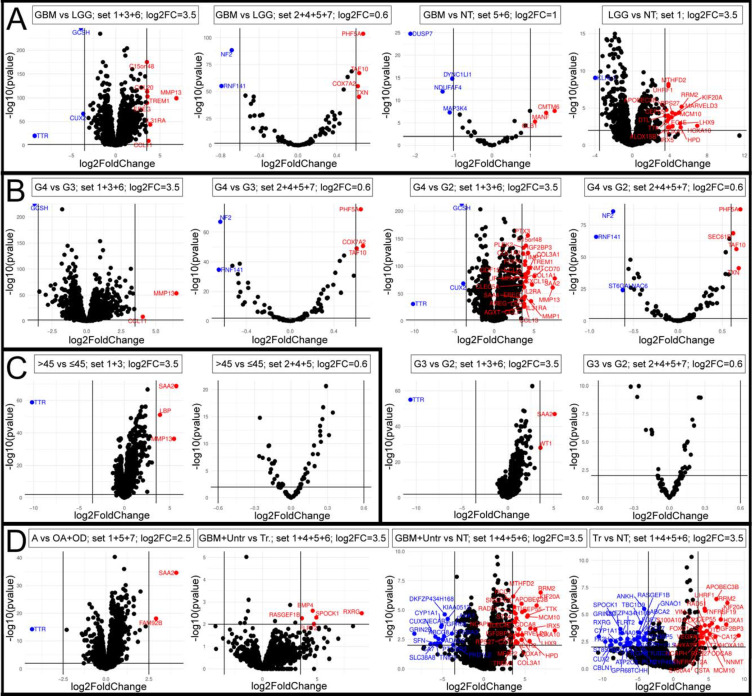
Differential gene expression analysis for each comparison. (**A**) Comparisons of *tumor_type*. (**B**) Comparisons of *neoplasm_histologic_grade*. (**C**) Comparisons of *age_at_initial_pathologic_diagnosis.* (**D**) Comparisons of *histological_type*. Genes marked with blue are downregulated while those in red are upregulated.

**Figure 5 cancers-13-02955-f005:**
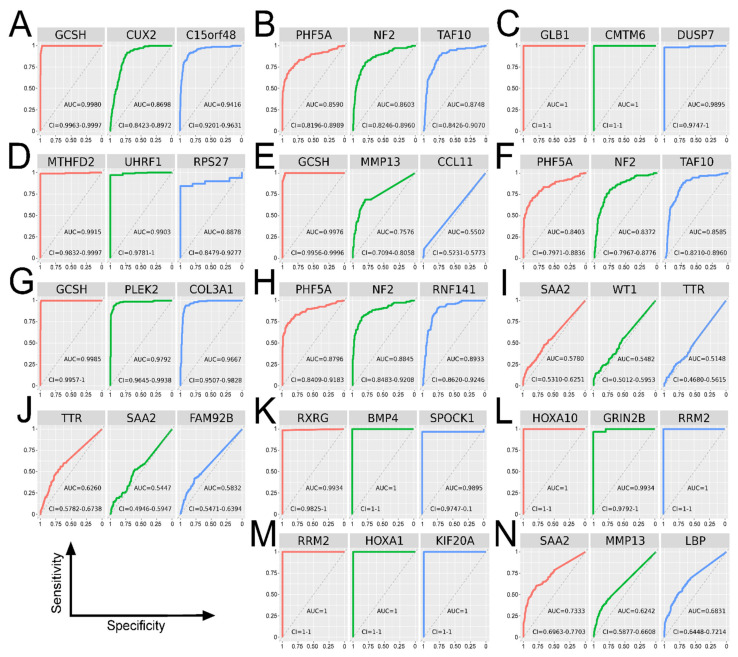
ROC analysis of top genes from comparisons—RNA-seq data. (**A**) GBM vs. LGG, set 1 + 3 + 6. (**B**) GBM vs. LGG, set 2 + 4 + 5 + 7. (**C**) GBM vs. NT, set 5 + 6. (**D**) LGG vs. NT, set 1. (**E**) G4 vs. G3, set 1 + 3 + 6. (**F**) G4 vs. G3, set 2 + 4 + 5 + 7. (**G**) G4 vs. G2, set 1 + 3 + 6. (**H**) G4 vs. G2, set 2 + 4 + 5 + 7. (**I**) G3 vs. G2, set 1 + 3 + 6. (**J**) A vs. OA and OD, set 1 + 5 + 7. (**K**) Untr GBM vs. Tr, set 1 + 4 + 5 + 6. (**L**) Untr GBM vs. NT, set 1 + 4 + 5 + 6. (**M**) Tr vs. NT, set 1 + 4 + 5 + 6. (**N**) >45 vs. ≤45, set 1 + 3.

**Figure 6 cancers-13-02955-f006:**
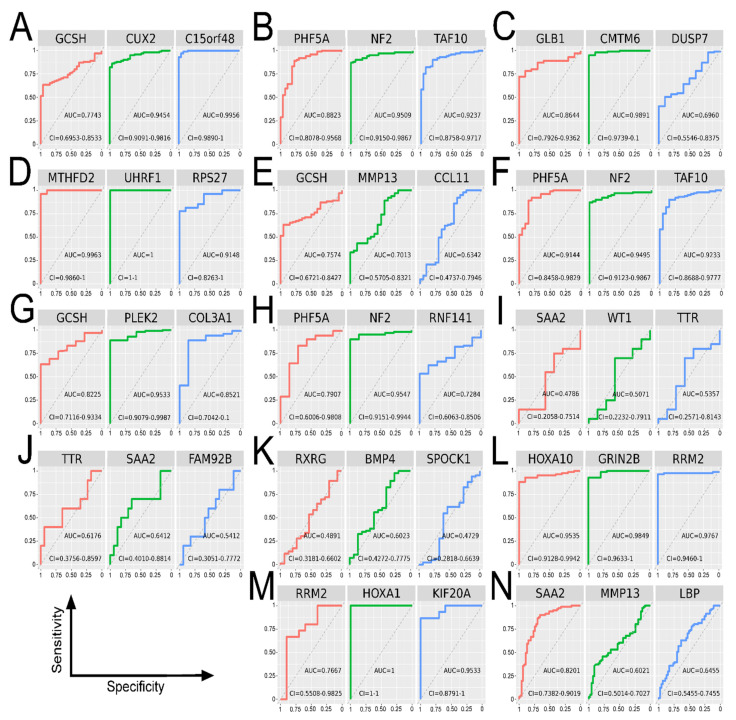
ROC analysis of top genes from comparisons—validation of the results using Agilent 244K G4502A microarray data. (**A**) GBM vs. LGG, set 1 + 3 + 6. (**B**) GBM vs. LGG, set 2 + 4 + 5 + 7. (**C**) GBM vs. NT, set 5 + 6. (**D**) LGG vs. NT set 1. (**E**) G4 vs. G3, set 1 + 3 + 6. (**F**) G4 vs. G3, set 2 + 4 + 5 + 7. (**G**) G4 vs. G2, set 1 + 3 + 6. (**H**) G4 vs. G2, set 2 + 4 + 5 + 7. (**I**) G3 vs. G2, set 1 + 3 + 6. (**J**) A vs. OA and OD, set 1 + 5 + 7. (**K**) Untr GBM vs. Tr, set 1 + 4 + 5 + 6. (**L**) Untr GBM vs. NT, set 1 + 4 + 5 + 6. (**M**) Tr vs. NT, set 1 + 4 + 5 + 6. (**N**) >45 vs. ≤ 45, set 1 + 3.

**Figure 7 cancers-13-02955-f007:**
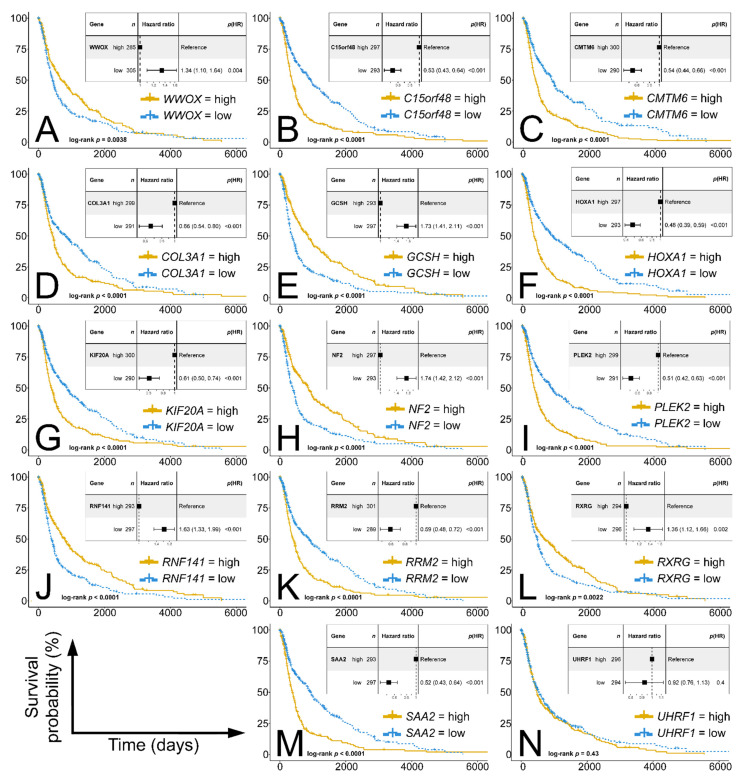
DFS analysis of the 13 most relevant genes and *WWOX.* (**A**) *WWOX*. (**B**) *C15orf48*. (**C**) *CMTM6*. (**D**) *COL3A1*. (**E**) *GCSH*. (**F**) *HOXA1*. (**G**) *KIF20A*. (**H**) *NF2*. (**I**) *PLEK2*. (**J**) *RNF141*. (**K**) *RRM2*. (**L**) *RXRG*. (**M**) *SAA2*. (**N**) *UHRF1*.

**Figure 8 cancers-13-02955-f008:**
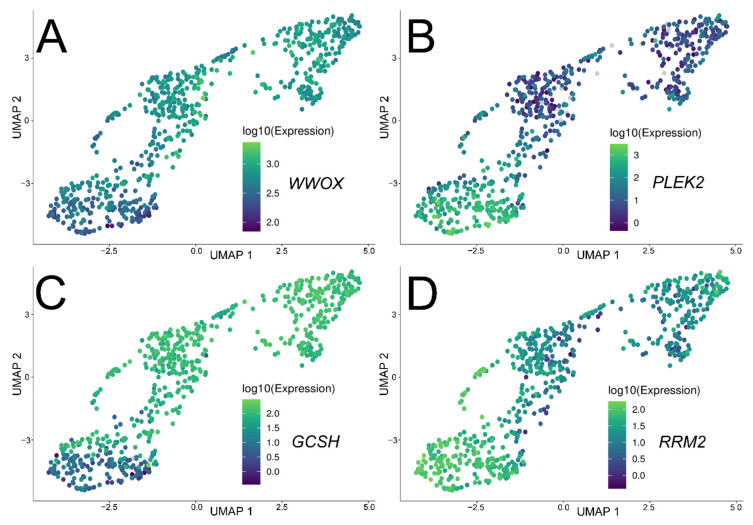
Spatial expression analysis of the 3 most relevant genes and *WWOX.* (**A**) *WWOX*. (**B**) *PLEK2*. (**C**) *GCSH*. (**D**) *RRM2*.

**Figure 9 cancers-13-02955-f009:**
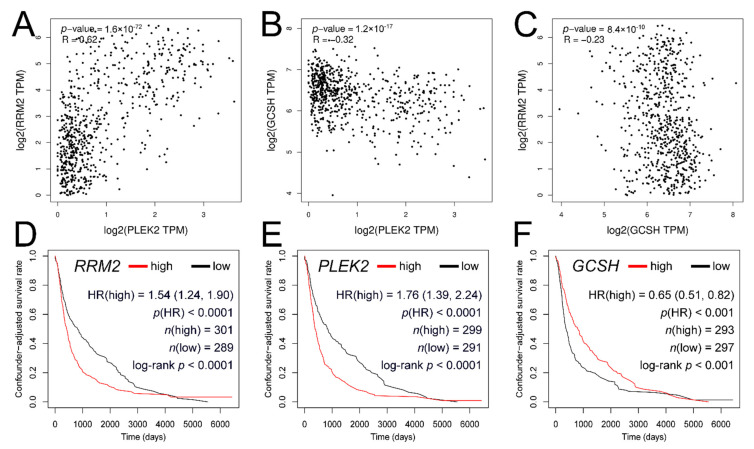
Correlation analysis of *PLEK2*, *RRM2*, and *GCSH* and its confounder-adjusted survival curves. (**A**) *PLEK2* vs. *RRM2*. (**B**) *PLEK2* vs. *GCSH*. (**C**) *GCSH* vs. *RRM2*. (**D**) ESTIMATE-adjusted survival for *RRM2*. (**E**) ESTIMATE-adjusted survival for *PLEK2*. (**F**) ESTIMATE-adjusted survival for *GCSH*.

**Figure 10 cancers-13-02955-f010:**
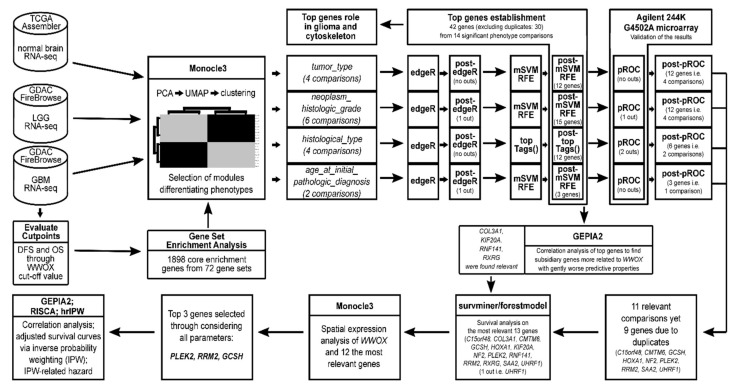
Methodology pipeline overview.

**Table 1 cancers-13-02955-t001:** The most relevant DEGs of each comparison appraised by mSVM-RFE or topTags().

Comparison	Set of Modules	Gene	AvgRank/Toptags
GBM vs. LGG	1 + 3 + 6	*GCSH*	1.0
		*CUX2*	2.2
		*C15orf48*	3.6
	2 + 4 + 5 + 7	*PHF5A*	1.0
		*NF2*	2.0
		*TAF10*	3.4
GBM vs. NT	5 + 6	*GLB1*	1.0
		*CMTM6*	2.3
		*DUSP7*	3.6
LGG vs. NT	1	*MTHFD2*	2.8
		*UHRF1*	3.3
		*RPS27*	4.7
G4 vs. G3	1 + 3 + 6	*GCSH*	1.0
		*MMP13*	2.0
		*CCL11*	3.0
	2 + 4 + 5 + 7	*PHF5A*	1.0
		*NF2*	2.0
		*TAF10*	3.3
G4 vs. G2	1 + 3 + 6	*GCSH*	1.8
		*PLEK2*	3.2
		*COL3A1*	5.8
	2 + 4 + 5 + 7	*PHF5A*	1.0
		*NF2*	2.1
		*RNF141*	2.9
G3 vs. G2	1 + 3 + 6	*SAA2*	1.1
		*WT1*	2.0
		*TTR*	2.9
A vs. OA + OD	1 + 5 + 7	*TTR*	First (topTags)
		*SAA2*	Second (topTags)
		*FAM92B*	Third (topTags)
Untr. GBM vs. Tr.	1 + 4 + 5 + 6	*RXRG*	First (topTags)
		*BMP4*	Second (topTags)
		*SPOCK1*	Third (topTags)
Untr. GBM vs. NT	1 + 4 + 5 + 6	*HOXA10*	First (topTags)
		*GRIN2B*	Second (topTags)
		*RRM2*	Third (topTags)
Tr. GBM vs. NT	1 + 4 + 5 + 6	*RRM2*	First (topTags)
		*HOXA1*	Second (topTags)
		*KIF20A*	Third (topTags)
>45 vs. ≤45	1 + 3	*SAA2*	1.0
		*MMP13*	2.1
		*LBP*	3.0

**Table 2 cancers-13-02955-t002:** Post-validation significant comparisons with the best explanatory gene.

Comparison	Set of Modules	Gene	AUC/95% CI/ACC
GBM vs. LGG	1 + 3 + 6	*C15orf48*	0.9686/0.9546–0.9816/0.9251
	2 + 4 + 5 + 7	*NF2*	0.9056/0.8698–0.9414/0.8591
GBM vs. NT	5 + 6	*CMTM6*	0.9946/0.9870–1/0.9805
LGG vs. NT	1	*UHRF1*	0.9952/0.9891–1/0.9875
G4 vs. G3	1 + 3 + 6	*GCSH*	0.8775/0.8339–0.9212/0.8255
	2 + 4 + 5 + 7	*NF2*	0.8934/0.8545–0.9322/0.8402
G4 vs. G2	1 + 3 + 6	*PLEK2*	0.9663/0.9362–0.9963/0.9255
	2 + 4 + 5 + 7	*NF2*	0.9196/0.8817–0.9576/0.8793
Untr. GBM vs. NT	1 + 4 + 5 + 6	*RRM2*	0.9884/0.9730–1/0.9861
Tr. GBM vs. NT	1 + 4 + 5 + 6	*HOXA1*	1/1–1/1
>45 vs. ≤45	1 + 3	*SAA2*	0.7767/0.7173–0.8361/0.7551

**Table 3 cancers-13-02955-t003:** Correlation analysis of the top genes with *WWOX*.

Gene	Correlation with *WWOX*
*BMP4*	R = 0.28, *p* < 0.0001
*C15orf48*	R = −0.37, *p* < 0.0001
*CCL11*	R = −0.13, *p* < 0.001
*CMTM6*	R = −0.21, *p* < 0.0001
*COL3A1*	R = −0.51, *p* < 0.0001
*CUX2*	R = 0.24, *p* < 0.0001
*DUSP7*	R = 0.28, *p* < 0.0001
*FAM92B*	R = −0.23, *p* < 0.0001
*GCSH*	R = 0.43, *p* < 0.0001
*GLB1*	R = −0.25, *p* < 0.0001
*GRIN2B*	R = 0.24, *p* < 0.0001
*HOXA1*	R = −0.26, *p* < 0.0001
*HOXA10*	R = −0.36, *p* < 0.0001
*KIF20A*	R = −0.42, *p* < 0.0001
*LBP*	R = −0.12, *p* < −0.0001
*MMP13*	R = −0.34, *p* < 0.0001
*MTHFD2*	R = 0.25, *p* < 0.0001
*NF2*	R = 0.20, *p* < 0.0001
*PHF5A*	R = −0.21, *p* < 0.0001
*PLEK2*	R = −0.44, *p* < 0.0001
*RNF141*	R = 0.58, *p* < 0.0001
*RPS27*	R = −0.11, *p* < 0.001
*RRM2*	R = −0.42, *p* < 0.0001
*RXRG*	R = 0.45, *p* < 0.0001
*SAA2*	R = −0.30, *p* < 0.0001
*SPOCK1*	R = 0.32, *p* < 0.0001
*TAF10*	R = −0.28, *p* < 0.0001
*TTR*	R = −0.033, *p* > 0.05
*UHRF1*	R = 0.078, *p* < 0.05
*WT1*	R = −0.13, *p* < 0.001

## Data Availability

Publicly available datasets were analyzed in this study. This data can be found here: https://gdac.broadinstitute.org/ and https://xenabrowser.net/ (accessed on 10 December 2020).

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
