# Peer review of "PLEK2*, *RRM2*, *GCSH*: A Novel *WWOX*-Dependent Biomarker Triad of Glioblastoma at the Crossroads of Cytoskeleton Reorganization and Metabolism Alterations"

_cancers, 2021, doi:10.3390/cancers13122955_

Round 1

Reviewer 1 Report

The research article from the authors attempted to elucidate the role of WW domain-containing Oxidoreductase (WWOX) dependent genes in glioblastoma, particularly as tumor suppressor in regulating cytoskeleton organization which resulted in glioma proliferation and invasiveness. However, to deliver such hypothesis we still find some confusing result and arising some question:

  • Figure 1. Please add information regarding patient number in the cohort of survival analysis
  • In methods part, subsection ‘Evaluation of statistical model accuracy’ the authors mentioned to use machine learning algorithm, however still does not discuss in detail what kind of algorithm that the authors used, whether it was neural network, decision tree, SVM, or etc must be provided in detail.
  • The flow of information to prove and conclude the assumed hypothesis is relatively unclear such as In Figure 2 the authors provide PCA to determine clinical factors which are worth to consider, however, we do not understand, what is the association with WWOX dependent genes in glioblastoma? Why the authors need to provide this information?
  • In result part, subsection ‘Differentially expressed genes were identified for each comparison of phenotypes’, the authors stated ‘DEGs were considered significant when at least log2FC = 0.6, however in many comparisons even log2FC = 3.5’, what does this mean? Why this sentence seems to be inconsistent about the log2FC criteria?
  • The research design which was using separation into several modules seems to be hard for the reader to digest, we suggest the authors need to explain this modules in more understandable fashion
  • Figure 7, we could not clearly see the hazard, if possible we suggest to change the plot into forest plot showing the hazard ratio of each significant genes
  • If the authors need to prove the prognostic model using WWOX-dependent genes, please provide Kaplan meier while impelementing the proposed model compared to standardized or established clinical scoring model in glioblastoma, this result may easily understandable by most other researcher
  • Glioblastoma has been already known to have several subtypes such as proneural, neural, classical, and mesenchymal. Could the authors explain the association of WWOX-dependent genes and tendency towards each subtype of glioblastoma?

From these essential issues above, we are reluctantly still in our position to reject this article. We hope in the future, the authors still try to submit other articles into our journal.

Author Response

Dear Reviewer 1,

Thank you so much for handling peer-review process.

Please see the attachment in order to download your Response Letter.

With kind regards and on behalf of all Authors,
Żaneta Kałuzińska

Reviewer 2 Report

Kaluzinska and collaborators used a combination of bioinformatic tools, data bases, web-based platforms, and machine learning algorithm to identify WWOX-dependent biomarkers in glioblastomas with a focus on genes involved in cytoskeleton dynamic. Thanks to this very interesting approach, they found that PLEK2, RRM2 and GCSH were differentially expressed between glioblastomas and low-grade gliomas.

Major revision

1) The way how glioblastoma cohorts were constructed is not clear. The cohorts should be the most homogenous as possible. Please verify that all the glioblastomas were IDH wild type glioblastoma because this is very not clear. A special paragraph in the methods part should be added to clarify it as well as all the clinical criteria that you used to selected also the other glioma subtypes.

2) The results should be validated at the protein level on independent cohorts of glioblastomas versus low grade gliomas

3) The discuss is way too long. Please rewrite it by being more concise and precise.

Minor revision

1) The term “glioblastoma multiform” is no longer right. Since the WHO classification 2016 (Louis et al, 2016) they are named glioblastoma IDH wt (the de novo glioblastoma). The secondary glioblastomas are now named glioblastoma IDH mutated. Please modify it in the test.

2) Please check the abstract because they are some excess space.

Author Response

Dear Reviewer 2,

Thank you so much for handling peer-review process.

Please see the attachment in order to download your Response Letter.

With kind regards and on behalf of all Authors,
Żaneta Kałuzińska

Reviewer 3 Report

In their manuscript entitled « PLEK2, RRM2, GCSH: a Novel WWOX-Dependent Biomarkers Triad of Glioblastoma at the Crossroads of Cytoskeleton Reorganization and Metabolism Alterations », the authors demonstrated, using public RNA-seq data and various bio-informatic tools, the impact of WWOX in glioblastoma pathogenicity through cytoskeleton remodelling and metabolic alterations. The work is nicely conducted. The results are impressive and open new roads to the field.

The font of some legend included into figures should be changed in order to be more readable.  

Author Response

Dear Reviewer 3,

Thank you so much for handling peer-review process.

Please see the attachment in order to download your Response Letter.

With kind regards and on behalf of all Authors,
Żaneta Kałuzińska

Round 2

Reviewer 2 Report

Thank you for this clear revised version